# Linac-Based Ultrahypofractionated Partial Breast Irradiation (APBI) in Low-Risk Breast Cancer: First Results of a Monoinstitutional Observational Analysis

**DOI:** 10.3390/cancers15041138

**Published:** 2023-02-10

**Authors:** Roland Merten, Mirko Fischer, Gennadii Kopytsia, Jörn Wichmann, Tim Lange, Anne Caroline Knöchelmann, Jan-Niklas Becker, Rüdiger Klapdor, Jan Hinrichs, Michael Bremer

**Affiliations:** 1Clinic for Radiotherapy, Hannover Medical School, 30625 Hannover, Germany; 2Clinic for Gynecology and Obstetrics, Hannover Medical School, 30625 Hannover, Germany; 3Institute for Radiology, Hannover Medical School, 30625 Hannover, Germany

**Keywords:** APBI, low-risk breast cancer, radiotherapy, hypofractionation

## Abstract

**Simple Summary:**

The subject of this trial was the introduction of adjuvant APBI. The radiation dose in the target volume and in organs at risk were evaluated retroactively, as was the acute toxicity. The exposure of the organs at risk was very low. Two out of three irradiated patients remained without any side effects. APBI is a very attractive treatment modality for patients with low-risk breast carcinoma.

**Abstract:**

**Purpose:** For adjuvant radiotherapy of low-risk breast cancer after breast-conserving surgery, there have been many trials of hypofractionation and partial breast irradiation (PBI) over the years, with proven mild long-term toxicity. The aim of this study was to introduce a short-course dose-adapted concept, proven in whole breast irradiation (WBI) for use in accelerated partial breast irradiation (APBI), while monitoring dosimetric data and toxicity. **Methods:** From April 2020 to March 2022, 61 patients with low-risk breast cancer or ductal carcinoma in situ (DCIS) were treated at a single institution with percutaneous APBI of 26 Gy in five fractions every other day after breast-conserving surgery. Dosimetric data for target volume and organs at risk were determined retrospectively. Acute toxicity was evaluated. **Results:** The target volume of radiotherapy comprised an average of 19% of the ipsilateral mamma. The burden on the heart and lungs was very low. The mean cardiac dose during irradiation of the left breast was only 0.6 Gy. Two out of three patients remained without any acute side effects. **Conclusions:** Linac-based APBI is an attractive treatment option for patients with low-risk breast cancer in whom neither WBI nor complete omission of radiotherapy appears to be an adequate alternative.

## 1. Introduction

Adjuvant radiotherapy of the breast has been an integral part of treatment after breast-conserving surgery for breast cancer for decades. Since its introduction, several large clinical trials have addressed hypofractionation schedules [1,2,3,4], and hypofractionation is now firmly established in routine clinical practice after initial concerns of increased late toxicity were refuted. Further studies have investigated the reduction of treatment volume in partial breast irradiation for suitable low-risk breast cancer patients [5,6,7,8]. The TARGIT-A [7] trial demonstrated high rates of local recurrence-free and low late toxicity for technically complex intraoperative radiotherapy, at least for patients with favorable risk. Strnad et al. [8] achieved comparable results with interstitial multi-catheter brachytherapy. Livi et al. [9] demonstrated, in the Florence study of percutaneous radiotherapy, that APBI with 30 Gy in five fractions had significantly lower late toxicity and no higher rate of local recurrence than in normofractionated whole breast irradiation (WBI), with an additional boost to the tumor bed, even after 10 years of follow-up [10], as confirmed by other long-term trials [5,11].

Exploring the appropriate fractionation schedule, balancing short radiation treatment time and long-term tolerability, the FAST-Forward trial [12] found slightly increased late toxicity in terms of fibrosis after whole breast irradiation (WBI) with 27 Gy in five fractions compared to 26 Gy in five fractions and 40 Gy in 15 fractions. Both dose concepts achieved comparably good local progression-free survival, even in the tumor bed as irradiation with 40 Gy in 15 fractions. At greater than a single dose of 5.2 Gy, the risk of fibrosis increases steeply. From the joint consideration of the Florence study [10] and the FAST-Forward study [12], it could be concluded that, in low-risk breast cancer, postoperative percutaneous APBI with 26 Gy in five fractions achieves the same freedom from recurrence both in the entire breast and in the tumor bed as normofractionated or moderately hypofractionated whole breast irradiation, and that acute and late toxicity is lower than with whole breast irradiation. APBI with 26 Gy in five fractions thus combines the best of the two worlds. Furthermore, the recent results of the Lumina study [13] underscored the need for further reduction of radiation treatment volume and overall treatment time in the low-risk setting, in which the treatment alternative to completely omitting postoperative radiotherapy might be partial breast irradiation, rather than whole breast irradiation.

The aim of our study was to quantitatively analyze dose-volume parameters of radiation treatment planning and to assess early normal tissue effects after the introduction of a linac-based, dose-adapted APBI.

## 2. Materials and Methods

From April 2020 to March 2022, 65 patients with newly diagnosed early breast cancer received postoperative APBI after breast-conserving surgery in our department and were retrospectively evaluated. All patients provided written informed consent in advance for blinded analysis of their clinical and treatment data. Additionally, a positive decision of the local ethical committee was achieved. Following the GEC-ESTRO guideline [5], the Florence trial [10], and the FAST-Forward trial [12], invasive carcinomas (*n* = 53) up to 30 mm in size or ductal carcinomas in situ (DCIS, *n* = 8) [14] up to 25 mm in size were allowed, as well as 40 mm in a single case of a patient pretreated with mantle field irradiation for Hodgkin’s disease years earlier. Patients had to be older than 50 years, with one exception made for the patient pretreated for Hodgkin’s disease. Thirty patients had left-sided breast irradiation, and 31 had right-sided breast irradiation. Further details of the patient population are shown in Table 1.

Staging classification was performed according to TNM 8th edition [15]. Resection with at least a 1 mm resection margin for invasive carcinoma and 2 mm for DCIS was needed. All but three of the carcinomas had a G 1–G 2 nonspecific type histology without expression of Her2neu. One patient with invasive lobular carcinoma underwent a preop MRI that excluded multicentricity. Forty-six patients (75% of all patients, 87% of invasive cancers) underwent histologic examination of axillary nodes, including 40 with sentinel node biopsy and six with axillary dissection. Eight patients with DCIS and seven patients with invasive carcinoma had no surgical staging of axillary lymphatic nodes because of a lack of consent from patients, but had unsuspicious findings (cN0) on sonography and CT scans. The absence of surgical axillary staging was not an exclusion criterion for our study. Sixty patients (98.4%) had no axillary LK metastases. One patient (1.6%) was treated with APBI, despite a singular axillary LK metastasis after neoadjuvant chemotherapy, and another aforementioned patient with invasive lobular carcinoma was treated with APBI at her own request. Two more patients with grade 3 tumors received chemotherapy and were treated with an APBI at their own request, although they were not completely suitable for partial breast irradiation according to current guidelines [16]. The following four patients had to be excluded from the study. Two had not undergone surgery. One had bilateral low-risk carcinoma with bilateral APBI. One withdrew her consent to participate in this analysis. A total of 61 patients were finally included in this analysis.

Radiotherapy started an average of 57 days after surgery (range 31–70 days). Radiotherapy was delivered using 6-MV linac photons with 26 Gy in five fractions, with one fraction every other day in most patients. The mean duration of the overall treatment time was 10 (range 5–14, median 10) days. Fifty-three patients were irradiated using the VMAT technique, and eight patients were irradiated using static fields after 3D planning. Three out of those eight patients were treated using the deep inspiration breathhold (DIBH) technique. During VMAT-Radiation, no DIBH was used.

All CT-based treatment planning (slide thickness: 3 mm) was performed using the treatment planning system Monaco (Elekta, Stockholm, Sweden) according to ICRU reports 50 and 62 [17,18] and ICRU report 83 [19], where appropriate. For definition of the clinical target volume (CTV), the former tumor bed was reconstructed on the basis of clinical information, surgical clips, preoperative imaging and radiopaque markers of the overlying skin scar. Preoperative mammography and sonography were available digitally for all patients, with additional preoperative MRI for nine patients and preoperative CT for five patients. The planning target volume (PTV) was the CTV, plus a margin of 5 mm in all directions, with at least 3 mm sparing of the skin surface where necessary.

The following structures were contoured as organs at risk: ipsilateral and contralateral breasts and both lungs, the heart, the LAD (left anterior descending artery), ipsilateral rib cage, and spinal cord.

Clinical examination was performed initially before treatment, at completion of radiotherapy, after 3 months and thereafter on a yearly schedule. Acute side effects were classified using CTCAE, version 5 [20].

Descriptive statistical analysis (average, whisker plot) was performed using the original Excel (Microsoft) spreadsheet, available as Appendix A.

## 3. Results

### 3.1. Quantitative Dose-Volume Analysis of PTV

The PTV was, on average, 110.5 cm^3^ (range: 43–538 cm^3^) and comprised, on average, 19.4% of the breast. The D_mean_ of PTV averaged 26.2 Gy (26.0 Gy for 3D and 25.9 Gy for VMAT). The V_90%_ averaged 21.7% of the ipsilateral breast, as shown in Figure 1.

Careful attention was paid to a homogenous dose distribution, due to the assumed marked increase in late tissue fibrosis beyond a total dose of 27 Gy [12]. Priority was given to avoiding a maximum of >105% of the prescribed dose. Therefore, the D_2%_ of PTV averaged 26.8 Gy, or 103.0% (range 100.4–105.6%). The D_98%_ averaged 23.9 Gy, or 92% of the prescribed dose (range 19.8 Gy–25.8 Gy, corresponding to 76.2% to 99.3% of the prescribed dose). The D_95%_ averaged 24.7 Gy. The global maximum averaged 27.2 Gy (27.4 Gy for the 3D-planned and 27.1 Gy for the VMAT-planned). Figure 2 illustrates the good homogeneity of the radiation dose in PTV.

The heterogeneity index was calculated according to ICRU report 83 [19]. The average heterogeneity index was 1.15 (range 1.09–1.25) for the 3D-planned patients and 1.08 (1.04–1.14) for the VMAT-planned patients.

The conformity index was calculated according to [21]. The mean conformity index was 0.66 (range 0.54–0.72, SD ± 0.07) for the 3D-planned patients and 0.85 (range 0.79–0.97, SD ± 0.09) for the VMAT-planned patients.

### 3.2. Organs at Risk

During APBI, very low radiation exposures were obtained for the contralateral breast and ipsilateral lung. As expected, a larger low dose volume was obtained after VMAT planning. After 3D planning, a larger high dose volume was suggestively recognizable. Further details are shown in Table 2.

For the contralateral lung, the average D_mean_ was 0.34 Gy (0.1 Gy for the 3D-irradiated and 0.39 Gy for the VMAT-irradiated). The V_5Gy_ was less than 1% in all cases.

For cardiac exposure, the overall doses were very low. The mean cardiac dose during irradiation of the left breast was only 0.6 Gy. A difference between the right and left irradiated groups was statistically found only for the 3D-irradiated group. There was no discernible side difference after VMAT planning. Further details are reported in Table 3.

For the ribs, the D_1mL_ averaged 22.7 Gy (range 9.1–26.5 Gy). The overall maximal dose for the ribs did not exceed 102% of the prescribed dose in any case.

### 3.3. Analysis of Acute Clinical Toxicity

The acute toxicity values were very low. Eight (13%) of the patients suffered from fatigue CTC 1°; 13 (20%) showed mild erythema of the irradiated skin corresponding to CTC 1°; and one patient with a superficially located target volume had dry desquamation corresponding to CTC 2°. One patient complained of tenderness in the irradiated breast. Forty of 61 patients (=66%) were without any acute toxicities. An overview of the overall toxicity is given in Figure 3. In the follow-up examination after 3 months, all acute toxicities had resolved in all patients. There was no prolonged acute toxicity. The mean follow-up was 4.8 months.

## 4. Discussion

Since partial breast irradiation (PBI) was shown to be oncologically equivalent to whole breast irradiation (WBI) in terms of local control in numerous trials for low-risk breast cancer, in the German interdisciplinary guideline [22], PBI is already approved as an established alternative to WBI in low-risk breast cancer. The ESTRO guideline [23] explicitly allows for APBI, in addition to moderate hypofractionated WBI, for use in routine care, even outside clinical trials. The 2021 AGO guideline [24] treatment options include PBI planned in moderate hypofractionation, as well as APBI in five fractions, delivered with the VMAT technique. The NSABP trial [11] performed APBI as single-catheter brachytherapy, including patients with limited (1–3) axillary lymph node metastases. Therefore, APBI is not recommended as a brachytherapy technique in the German AGO guideline (Working Group for Gynecological Oncology), due to a slightly increased rate of local recurrence. For these technical reasons, we did not consider brachytherapy for our patients. The RAPID trial [4] demonstrated an increased rate of fibrosis as late toxicity after APBI with 38.5 Gy in 10 fractions, two fractions per day. This high dosage for partial breast irradiation, with administration twice daily, has now been shown to be unnecessary: Brunt was able to show, in the 10-year follow-up of the FAST study [25] that, with equally good locoregional control, 28.5 Gy in five fractions did not cause more fibrosis than 50 Gy in 25 fractions, not even in WBI. The dose concept (5 × 6 Gy every other day) successfully used in the Florence study [10] was also significantly lower than the doses used in the RAPID study [4]. Thus, the initial concerns about an increased rate of fibrosis after APBI have been clearly refuted. The dose and fractionation from the FAST-Forward trial of 26 Gy in five fractions [12], which have been proven for WBI, were therefore adopted for our study, and the irradiation every other day as used in the Florence-trial [10] and the FAST trial [25]. The dose concept of our study has the advantages of APBI combined with the excellent tolerability in the FAST-Forward trial for patients [12]. The use of our dose and fractionation for APBI follows on logically from the trials mentioned, but is a novelty in radiotherapy. Currently, the long-term follow-up is underway to ensure that we also achieve long-term local control as well as in the cited trials. We will report on this in a later publication.

Brunt et al. [12] was also able to demonstrate that, in APBI after 10 years, 30 Gy in five fractions caused more fibrosis than 28.5 Gy in five fractions. Thus, it was shown that, within the narrow range of single doses between 5.2 and 5.7 Gy, there is a steep gradient of fibrosis development in whole breast irradiation, which should not be exceeded. Therefore, in our study, a D_2%_ of 103% of the reference dose was not exceeded, since avoidance of dose peaks was a well-founded priority. Nevertheless, ICRU-conforming [19] irradiation of the PTV was achieved in all cases.

The British consensus recommendation of the Royal College of Radiologists recommends PBI for the low-risk setting, but has thus far approved only elaborate multi-catheter brachytherapy for APBI [8,26]. The British NICE guideline [27] already recommends percutaneous radiotherapy (EBRT) for PBI. A critical discussion of the different techniques for APBI was published by Njeh et al. [28] in 2010. With the use of VMAT technology, we follow these recommendations on PBI. In terms of interplay effect, any percutaneous radiotherapy is inferior to interstitial brachytherapy. This technical detail could be neglected in the large clinical trials on PBI without any clinically tangible disadvantage [4,9]. Since we do not have breath gating, we have refrained from using the flattening filter free (FFF) technique. Intraoperative single-shot brachytherapy in the TARGIT-A trial [7] achieved good local control with low toxicity in the 10-year follow-up, but has not become a routine procedure because of the high expense and perioperative complications. Crucial to the good oncologic outcomes of APBI is the careful selection of suitable patients. The inclusion of lymph node-positive or invasive lobular patients led to increased local recurrences (ELIOT-Trial) [6] and was therefore avoided in the present study, except for the two exceptions mentioned above, who were treated with APBI at their own request and whose statistical evaluation for dosimetry and acute toxicity did not pose a problem. Not starting radiation therapy until 8 weeks after surgery is due to capacity and waiting times in our region. There was no preferential treatment for the patients in our retrospective study. This is real-world data.

Since the 3D technique was a long-standing practice for WBI, APBI was initially started in the 3D technique, but was soon switched to the VMAT technique to take advantage of the superior conformity and homogeneity of this technique. Figure 2 shows that the recommendations of the ICRU were followed, despite the change of methods, especially with regard to the avoidance of dose maxima.

Radiation exposure to the heart, contralateral breast and lung is slightly higher with the VMAT technique than with the 3D technique, but is far less than the critical limits [29,30,31]. Overall, because of the small size of the PTV, the technical advantages of the VMAT technique almost always predominate in the ipsilateral breast. Other studies have also been able to demonstrate this outcome in methodological comparisons [32,33]. The results of radiation exposure to the lung were excellent in APBI, far less than the established constraints in all cases [34,35], which are occasionally difficult to meet in WBI [36]. Pulmonary toxicity accordingly did not occur in our study, consistent with reports from other studies [37,38]. Dry desquamation occurred in only one case of superficial PTV. The presented results confirmed the results of other studies [39,40] after adoption of this therapy concept without any problems. The good tolerability of APBI makes this form of radiotherapy an attractive treatment option for low-risk breast cancer if one wishes to avoid the somewhat increased risk of local recurrence that is imminent if radiotherapy is omitted [41,42]. However, especially in cases of significantly reduced life expectancy, foregoing radiotherapy may also be the gentle alternative.

The limitations of our trial are the small number of cases and the retrospective collection of data. Due to the comparatively short follow-up, the risk for the development of radiogenic fibrosis cannot yet be conclusively assessed.

## 5. Conclusions

Our early experience with linac-based APBI presented here confirmed the low acute toxicity of APBI reported by others [43,44]. Sixty-six percent of irradiated patients remained without any side effects. Because of its excellent tolerability, APBI should be used consistently in defined low-risk situations to further reduce the treatment burden for suitable patients with low-risk breast cancer. For this selected patient population with low-risk breast cancer, APBI represents an attractive treatment option in which neither WBI nor complete omission of radiotherapy [13] appears to be an adequate alternative.

## Figures and Tables

**Figure 1 cancers-15-01138-f001:**
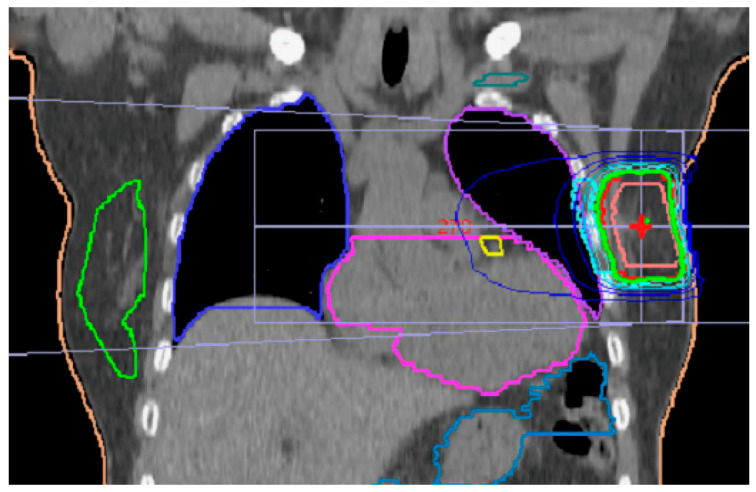
On average, 21% of the breast were irradiated by 90% of prescribed dose. Green isodose represents 95%. Yellow mark is LAD (left anterior descending artery), which was contoured in all left-side patients. Teal mark is for brachial nerve plexus. Contralateral breast is marked green.

**Figure 2 cancers-15-01138-f002:**
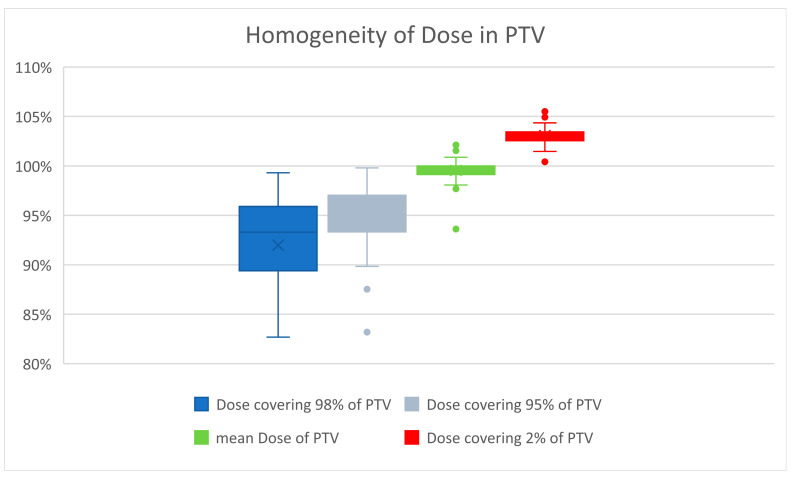
Dose maxima greater than 105% were carefully avoided because of the higher risk of developing fibrosis. The mean dose averaged 100%. The small boxes and short whiskers illustrate the consistent quality of treatment planning in patients.

**Figure 3 cancers-15-01138-f003:**
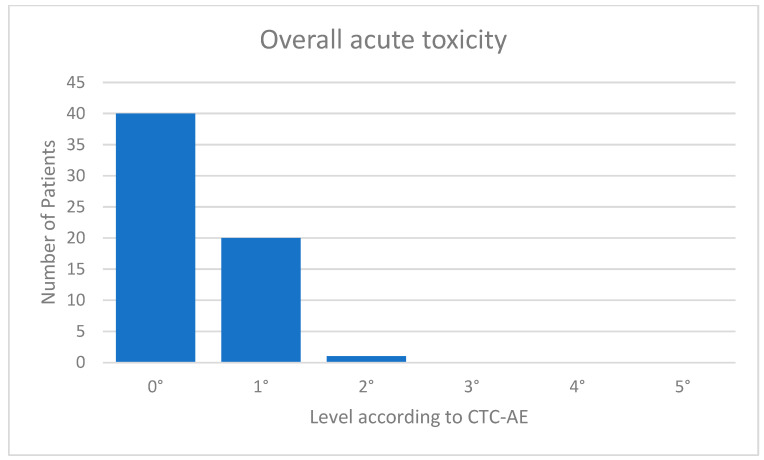
Acute toxicity.

**Table 1 cancers-15-01138-t001:** Patients‘ properties.

T-Stage				
T i.s. *n* = 8	T1 *n* = 43	T2 *n* = 10	T3 *n* = 0	T4 *n* = 0
Location quadrant of the breast				
Upper outer quadrant *n* = 34	Upper inner quadrant *n* = 14	Lower outer quadrant *n* = 5	Lower inner quadrant *n* = 5	Central *n* = 3

Age mean 68 years (range: 48–86 years.).

**Table 2 cancers-15-01138-t002:** Very low radiation doses were shown for the ipsilateral lung and contralateral breast.

		All	3D-Planned	VMAT
Lung ipsilateral	V_5Gy_	7.5%	5.2%	7.9%
	V_10Gy_	1.8%	1.5%	1.8%
	V_15Gy_	0.6%	0.7%	0.6%
	V_20Gy_	0.2%	0.4%	0.1%
Breast contralateral	D_mean_	0.32 Gy	0.11 Gy	0.36 Gy
	D_2%_	1.06 Gy	0.29 Gy	1.19 Gy

**Table 3 cancers-15-01138-t003:** Mean dose (and range) of radiation to the heart [Gy].

	Right Breast Irradiated	Left Breast Irradiated
3D planning (*n* = 8)	0.13 (0.07–0.19)	0.44 (0.21–0.65)
VMAT planning (*n* = 53)	0.57 (0.14–1.65)	0.67 (0.15–2.1)
All	0.51 (0.07–1.65)	0.63 (0.07–2.1)

## Data Availability

Research data are stored in an institutional repository and will be shared upon request to the corresponding author.

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
