# Peer review of "Linac-Based Ultrahypofractionated Partial Breast Irradiation (APBI) in Low-Risk Breast Cancer: First Results of a Monoinstitutional Observational Analysis"

_cancers, 2023, doi:10.3390/cancers15041138_

Round 1
Reviewer 1 Report
The authors presented a retrospective analysis of 61 patients with low-risk breast cancer, particularly through assessing the treatment planning outcomes in an effort to evaluate early normal tissue effects following a linac-based dose adapted APBI.
The manuscript is well-written, with an excellent introductory and discussion sections. The subject discussed is highly topical and covers an essential subject for the radiation oncology community. Nonetheless, the manuscript suffers from minor flaws that the authors must address prior to the publication.
1) Results was poorly described. Can you please summarise the results in table for a better visualisation. Particularly, the dose-volume analysis of PTV along with the OARs values.
2) Was DIBH approach adopted? Please clarify.
3) The discussion currently reads as a review of what other studies have reported, which is interesting and important, but they are not clearly linked back to the results.
4) Please include in the discussion section a paragraph about the interplay effect. VMAT irradiates only a portion of the target volume at a certain time. This creates the possibility of significant dosimetric missing of the target volume, which may, in turn, have an undesirable influence on local tumour control. This phenomenon has been recognised as ‘interplay effect’.
5) Radiobiological perspective of APBI is missing. For instance, prolonged total treatment time, postponed irradiation, and the possibility of accelerated repopulation of tumour cells can adversely affect local tumour control. Please discuss accelerated repopulation in depth from a radiobiological perspective.
Minor issues;
1) Summary (lines 12-13), please re-write this section for an additional clarity (too many ands)
2) Abstract: Line 24; Too many ands
3) Introduction Line 43; Please define APBI
4) Conclusion: No need to put reference in the conclusion section as this has been displayed in several sections of the manuscript
Author Response
1) Results was poorly described. Can you please summarise the results in table for a better visualisation. Particularly, the dose-volume analysis of PTV along with the OARs values.
Thank you for this valuable and constructive advice given by all reviewers. I inserted a table and a figure for better illustration.
2) Was DIBH approach adopted? Please clarify.
DIBH was not used with 3 exceptions. I have explained this in the chapter "Material and Methods".
3) The discussion currently reads as a review of what other studies have reported, which is interesting and important, but they are not clearly linked back to the results.
I have made the close links between our results and the cited studies more explicit in the discussion.
4) Please include in the discussion section a paragraph about the interplay effect. VMAT irradiates only a portion of the target volume at a certain time. This creates the possibility of significant dosimetric missing of the target volume, which may, in turn, have an undesirable influence on local tumour control. This phenomenon has been recognised as ‘interplay effect’.
At your suggestion, I mentioned the interplay effect in the discussion. We are taking it into account in FFF stereotactic radiosurgery of the heart in the RAVENTA trial in which we are participating. In radiotherapy of breast cancer, it has not been mentioned in any of the large clinical trials so far.
5) Radiobiological perspective of APBI is missing. For instance, prolonged total treatment time, postponed irradiation, and the possibility of accelerated repopulation of tumour cells can adversely affect local tumour control. Please discuss accelerated repopulation in depth from a radiobiological perspective.
I have added radiobiological aspects to the discussion. However, we have not made any fundamental changes to the proven concepts of fractionation and therefore do not expect any relevant differences in the efficacy of our therapy.
Minor issues;
1) Summary (lines 12-13), please re-write this section for an additional clarity (too many ands)
I used a better wording, now.
2) Abstract: Line 24; Too many ands
I used a better wording, now.
3) Introduction Line 43; Please define APBI
APBI is already defined in the heading and in line 20.
4) Conclusion: No need to put reference in the conclusion section as this has been displayed in several sections of the manuscript
To show the close connection of our conclusion with the already published trials, I would like to mention references in this section as well. Nobody is forced to pay attention to them, just in case of interest in details.
best regards
Roland Merten
Reviewer 2 Report
Overall, this paper has been well written.
However, some remarks need to be resolved:
- The title is not clear on the methodology, which is a pure observational reporting, in case only on acute toxicity. This could be more precise.
- In the introduction, the authors refer for APBI to the Targit trial (with beneficial findings) although this trial has been subject to strong doubts on the 'statistic fitting of data' (cf. comment Cuzick). The Eliot trial had negative findings but a much more robust methodology.
- The authors refer to the Fast Forward trial for their prescription of 5x5.2Gy, but from the methodology it appears they combine this prescription with the HAI5 5x5.7Gy every other day. This modification has not been discussed or motivated.
- The results section provides a long narrative of volumetric data: reading may improve if these 'quantative data' would have been reported in a table.
- The place of PBI versus omission of radiotherapy (with no radiation toxicity at all) is not discussed. Nevertheless this seems the most relevant discussion in EB-PBI for low risk breast cancer. Although the authors touch this topic briefly in their discussion, a more thorough discussion comparing PBI with the PRIME and CALGB9343 trial touches the core of the relevance of EB-PBI and should be elaborated.
Author Response
- The title is not clear on the methodology, which is a pure observational reporting, in case only on acute toxicity. This could be more precise.
Thank you very much for your helpful and constructive comments. I have made the title more precise.
- In the introduction, the authors refer for APBI to the Targit trial (with beneficial findings) although this trial has been subject to strong doubts on the 'statistic fitting of data' (cf. comment Cuzick). The Eliot trial had negative findings but a much more robust methodology.
I have now mentioned that the TARGIT-trial is controversial in its statistical analysis. We participated in the TARGIT-trial at that time, but now do not practice intraoperative irradiation because of the serious difficulties with lengthy seromas.
- The authors refer to the Fast Forward trial for their prescription of 5x5.2Gy, but from the methodology it appears they combine this prescription with the HAI5 5x5.7Gy every other day. This modification has not been discussed or motivated.
I have now better justified the selection of our fractionation in the discussion. This radiobiological aspect has also been criticized by another reviewer.
- The results section provides a long narrative of volumetric data: reading may improve if these 'quantative data' would have been reported in a table.
Thank you very much for your comment. All reviewers have criticized this point. I have therefore added a table and an illustration.
- The place of PBI versus omission of radiotherapy (with no radiation toxicity at all) is not discussed. Nevertheless this seems the most relevant discussion in EB-PBI for low risk breast cancer. Although the authors touch this topic briefly in their discussion, a more thorough discussion comparing PBI with the PRIME and CALGB9343 trial touches the core of the relevance of EB-PBI and should be elaborated.
Thank you for your reference to those two trials. I was previously only aware of the Lumina study from 2022. I have added this point to the discussion. In fact, in clinical practice, we already occasionally omit radiotherapy in very old patients.
best regards
Roland Merten
Reviewer 3 Report
The authors report their experience in the use of accelerated partial breast irradiation in patients with low-risk breast cancer after breast-conserving surgery. The authors evaluate the side effects of the fractionation used in the study. The manuscript is interesting even if it needs some integration.
Lines 22, 67, 78. I advise the authors to verify the total number of patients enrolled and treated in the study. It seems to me that there is no correspondence between the numbers reported.
Line 102 I wonder why the authors treated 3 patients with 3D technique and if they can specify the reasons in the study.
Line 121. I advise the authors to report the data of the quantitative analysis of PTV and organs at risk in a Table. At least the most important ones as they have already done for the heart.
The study can be re-evaluated after this Minor Revision.
Author Response
The authors report their experience in the use of accelerated partial breast irradiation in patients with low-risk breast cancer after breast-conserving surgery. The authors evaluate the side effects of the fractionation used in the study. The manuscript is interesting even if it needs some integration.
Lines 22, 67, 78. I advise the authors to verify the total number of patients enrolled and treated in the study. It seems to me that there is no correspondence between the numbers reported.
Thank you very much for your constructive comments. All numbers fit together. Unfortunately, we were able to include only 61 of the irradiated patients in the statistical evaluation at last count.
Line 102 I wonder why the authors treated 3 patients with 3D technique and if they can specify the reasons in the study.
As described in the discussion, we first started the APBI in 3D-technology and soon switched to VMAT-technology. Our study is retrospective and does not have a prospectively determined technique. However, we were clear about the inclusion criteria when we introduced APBI.
Line 121. I advise the authors to report the data of the quantitative analysis of PTV and organs at risk in a Table. At least the most important ones as they have already done for the heart.
Many thanks for this advice, which all reviewers have mentioned. I have added a table and a figure to show the results more clearly.
The study can be re-evaluated after this Minor Revision.
Reviewer 4 Report
Summary: this study essentially serves as a 61-patient data supplement from a single center, to the large, 4,100 patient UK FAST-FORWARD clinical trial with a focus of PBI. The latter trial describes similar outcomes in terms of tumor control and cosmesis by delivering 26 Gy in 5 fx’s (5.2 Gy/fx), as compared to Canadian or UK standard hypofractionation of 40.05 Gy in 15 fx’s (2.67 Gy/fx). While contribution of this data and confirmation of an existing results/reproducibility in a clinical trial certainly contributes valuable information, and differs from the original FAST-FORWARD trial in the sense that this examines partial-breast specifically, the study as it stands alone lacks novelty, and unless a more detailed and novel breakdown of the data can be added, it is recommended that this work be contributed as a supplement or a Case Study. For example, the following two works also examine partial breast, accelerated hypo-fractionation:
Al-Rashdan, A., Roumeliotis, M., Quirk, S., Grendarova, P., Phan, T., Cao, J., ... & Barbera, L. (2020). Adapting radiation therapy treatments for patients with breast cancer during the COVID-19 pandemic: hypo-fractionation and accelerated partial breast irradiation to address World Health Organization recommendations. Advances in Radiation Oncology, 5(4), 575-576.
Shaitelman, S. F., Khan, A. J., Woodward, W. A., Arthur, D. W., Cuttino, L. W., Bloom, E. S., ... & Vicini, F. A. (2014). Shortened Radiation Therapy Schedules for Early‐Stage Breast Cancer: A Review of Hypofractionated Whole‐Breast Irradiation and Accelerated Partial Breast Irradiation. The breast journal, 20(2), 131-146.
fractionation: whereas the former was submitted as a “letter to the editor” to Advances in Radiation Oncology, and not an original research manuscript. This could serve as a viable alternative.
Abstract
The details of the hypofractionation scheme could be made clearer in the abstract, as it is not later stated until the introduction that all five fractions are delivered in the same week, as per the FAST Forward trial (i.e. as opposed to other five-fraction schemes such as five fractions once per week over five weeks).
Introduction:
- Overall, a clear introduction providing description of the results of previous seminal works.
- May be reasonable to add detail to definitions of low-risk breast carcinoma, given that the works’ conclusions surround namely low-risk disease.
Methods
- Line 118, include details on which statistical tests were performed in Excel.
Results:
- How was the heterogeneity index calculated? May need to bring in/supplement with a bit more information here.
- I believe this work could be supplemented with more informative graphics, and the lack thereof is a weak point of the manuscript. There appears to be substantial and quality data to present but the work presents only one bar chart illustrating number of patients’ overall degree of toxicity, but otherwise lacks meaningful representation of other data, volumes, specific OAR’s etc. The work also lacks an analysis of error and a thorough statistical analysis.
- The combination of 3D and VMAT PBI treatments may convolute the conclusions this works aims to draw around fibrosis induced/tumor control by comparison of fractionation scheme alone. It may be of interest (or at least a reasonable expression of quality control) to prove that the dose control was volumetrically similar (hot spot and coverage volumes) between both planning modalities.
Conclusions:
- The conclusions are fair and not over-reaching, but again find this work to serve more as a useful supplement to existing works than it does as an original research manuscript.
Author Response
Summary: this study essentially serves as a 61-patient data supplement from a single center, to the large, 4,100 patient UK FAST-FORWARD clinical trial with a focus of PBI. The latter trial describes similar outcomes in terms of tumor control and cosmesis by delivering 26 Gy in 5 fx’s (5.2 Gy/fx), as compared to Canadian or UK standard hypofractionation of 40.05 Gy in 15 fx’s (2.67 Gy/fx). While contribution of this data and confirmation of an existing results/reproducibility in a clinical trial certainly contributes valuable information, and differs from the original FAST-FORWARD trial in the sense that this examines partial-breast specifically, the study as it stands alone lacks novelty, and unless a more detailed and novel breakdown of the data can be added, it is recommended that this work be contributed as a supplement or a Case Study. For example, the following two works also examine partial breast, accelerated hypo-fractionation:
Al-Rashdan, A., Roumeliotis, M., Quirk, S., Grendarova, P., Phan, T., Cao, J., ... & Barbera, L. (2020). Adapting radiation therapy treatments for patients with breast cancer during the COVID-19 pandemic: hypo-fractionation and accelerated partial breast irradiation to address World Health Organization recommendations. Advances in Radiation Oncology, 5(4), 575-576.
Shaitelman, S. F., Khan, A. J., Woodward, W. A., Arthur, D. W., Cuttino, L. W., Bloom, E. S., ... & Vicini, F. A. (2014). Shortened Radiation Therapy Schedules for Early‐Stage Breast Cancer: A Review of Hypofractionated Whole‐Breast Irradiation and Accelerated Partial Breast Irradiation. The breast journal, 20(2), 131-146.
An APBI in the fractionation we used has not been described in any of the studies mentioned. We have taken the fractionation from the UK-FAST study by Brunt et al., who used it for WBI. While this is not a revolution in radiation therapy, it is a careful retrospective analysis that is well worthy as an original article in a scientific journal. It is a novelty in radiation oncology.
fractionation: whereas the former was submitted as a “letter to the editor” to Advances in Radiation Oncology, and not an original research manuscript. This could serve as a viable alternative.
Our study is methodologically too elaborate and too extensive to be presented as a letter to the editor.
Abstract
The details of the hypofractionation scheme could be made clearer in the abstract, as it is not later stated until the introduction that all five fractions are delivered in the same week, as per the FAST Forward trial (i.e. as opposed to other five-fraction schemes such as five fractions once per week over five weeks).
We have now clearly described the fractionation concept we use, as this is our novelty. It is an intellectual fusion of the Florence-trial and the Fast-trial.
Introduction:
- Overall, a clear introduction providing description of the results of previous seminal works.
May be reasonable to add detail to definitions of low-risk breast carcinoma, given that the works’ conclusions surround namely low-risk disease.
Thank you very much for this helpful and constructive comment. We have described our inclusion criteria more precisely in some places. This hint was also given by another reviewer.
Methods
- Line 118, include details on which statistical tests were performed in Excel.
I have added that.
Results:
How was the heterogeneity index calculated? May need to bring in/supplement with a bit more information here.
The heterogeneity index is calculated as recommended by the ICRU:
(D2%-D98%)/prescribed Dose. As a result you get e.g. 0.12. The bigger the number, the worse is your plan. In our software (Monaco by Elekta) for treatment planning it is automatically displayed for each plan and we take it into account when checking the plans. Every days work.
I believe this work could be supplemented with more informative graphics, and the lack thereof is a weak point of the manuscript. There appears to be substantial and quality data to present but the work presents only one bar chart illustrating number of patients’ overall degree of toxicity, but otherwise lacks meaningful representation of other data, volumes, specific OAR’s etc. The work also lacks an analysis of error and a thorough statistical analysis.
Thank you very much for this hint, which all reviewers have attached. We have now presented the results more clearly with an additional table and a figure. The supplemented Whisker-plot also shows the possible errors.
The combination of 3D and VMAT PBI treatments may convolute the conclusions this works aims to draw around fibrosis induced/tumor control by comparison of fractionation scheme alone. It may be of interest (or at least a reasonable expression of quality control) to prove that the dose control was volumetrically similar (hot spot and coverage volumes) between both planning modalities.
Thank you, we explained that in the discussion and made it clear in the illustration.
Conclusions:
The conclusions are fair and not over-reaching, but again find this work to serve more as a useful supplement to existing works than it does as an original research manuscript.
This fractionation for APBI is our novelty. It is an intellectual fusion of the Florence-trial and the Fast-trial.
Reviewer 5 Report
General comment: the role of APBI is still controversial and discussed. Limitations of this report but also in general of literature on this topic is frequently the retrospective nature of the trials and the population as well as techniques heterogeneity.
1) please describe low risk parameters in particular explain if there are age limits (age > 50 ?)
2) please insert a table with patients characteristics
3) please clarify what patients are considered eligible for analysis; lines 89-97 are not clear
4) Line 167: what's the meaning of "oncologically equivalent"?
Author Response
General comment: the role of APBI is still controversial and discussed. Limitations of this report but also in general of literature on this topic is frequently the retrospective nature of the trials and the population as well as techniques heterogeneity.
Yes!! APBI is also much discussed among gynecologists and radiotherapists in our city. That's why we performed our analysis after introducing APBI in Hannover Medical School.
1) please describe low risk parameters in particular explain if there are age limits (age > 50 ?)
We selected strict inclusion criteria from the known studies of the APBI, which we adhere to at our clinic. Thank you for your comment, I have clarified the description in the manuscript.
2) please insert a table with patients characteristics
Thank you for your comment. We have presented the patients' properties in a table that also describes the age of the patients.
3) please clarify what patients are considered eligible for analysis; lines 89-97 are not clear
Thank you. That needed to be made clear.
4) Line 167: what's the meaning of "oncologically equivalent"?
Thank you. I have made that clear.
Round 2
Reviewer 1 Report
Very interesting topic, and hence, the article is worth to be published in its current form.